# A Systematic Review of UK Educational and Training Materials Aimed at Health and Social Care Staff about Providing Appropriate Services for LGBT+ People

**DOI:** 10.3390/ijerph16244976

**Published:** 2019-12-07

**Authors:** Ros Hunt, Christopher Bates, Susan Walker, Jeffrey Grierson, Sarah Redsell, Catherine Meads

**Affiliations:** Faculty of Health, Education, Medicine and Social Care, Anglia Ruskin University, East Road, Cambridge CB1 1PT, UK; ros.hunt@virgin.net (R.H.); Chrisbates1809@gmail.com (C.B.); susan.walker@anglia.ac.uk (S.W.); jeffrey.grierson@anglia.ac.uk (J.G.); sarah.redsell@anglia.ac.uk (S.R.)

**Keywords:** lesbian, bisexual, gay, transgender, education, systematic review

## Abstract

Background: There is greater dissatisfaction with health services by LGBT people compared to heterosexual and cisgender people and some of this is from lack of equality and diversity training for health professionals. Core training standards in sexual orientation for health professionals have been available since 2006. The purpose of this project is to systematically review educational materials for health and social care professionals in lesbian, gay, bisexual, and transgender (LGBT) issues. Methods: A protocol was developed and searches conducted in six databases. Selection criteria: any studies reporting delivery or evaluation of UK education of health and/or social care professionals in LGBT issues, with no language or setting restrictions. Inclusions and data extraction were conducted in duplicate. Narrative synthesis of educational evaluations was used. Educational materials were assessed using thematic synthesis. Results: From the searches, 165 full papers were evaluated and 19 studies were included in the narrative synthesis. Three were successful action-research projects in cancer services and in residential care. Sixteen sets of educational/training materials have been available since 2010. These varied in length, scope, target audience, and extent of development as classroom-ready materials. Conclusions: Despite the availability of appropriate training programmes for post-qualifying staff, recommendations to undertake training, best practice examples, and statements of good intent, LGBT people continue to report that they are experiencing discrimination or direct prejudice from health and/or social care services. Better training strategies using behaviour change techniques are needed.

## 1. Introduction

The recent survey of 108,100 lesbian, gay, bisexual, and transgender (LGBT) people’s experiences of everyday life in the UK, published by the UK Government Equalities Office [1] found that in the preceding 12 months 40% of transgender respondents had had at least one negative experience of healthcare because of their gender identity and that 13% of cisgender respondents had had at least one negative experience of healthcare because of their sexual orientation. This finding echoes that of a recent review of the literature in inequality among LGBT groups in the UK [2] which found greater dissatisfaction with health services by LGBT people compared to heterosexual and cisgender people. They report instances of overt discrimination and inappropriate behaviour by health professionals from a number of different studies included in the review.

A recent grey literature survey of 3000 UK health and social care staff (called the Unhealthy Attitudes Survey) [3] found that 25% of patient-or client-facing staff had heard colleagues make negative remarks about sexual orientation and 20% make negative remarks about gender identity. Five percent of patient-or client-facing staff had witnessed colleagues discriminate against or provide poorer treatment because of their sexual orientation and 7% of health and social care staff said they would not feel comfortable working alongside a trans colleague. Numerous quotes are provided, such as “A colleague who is gay made a remark about his partner and another colleague said ‘Oh my god seriously are you gay, gross’. The irony of this was that the remark was made during equality and diversity training.” and “A transgender nurse [was] often referred to as ‘he-she-it’ by other staff and service users” [3]. Staff remarks to other staff have the potential to more clearly indicate their attitudes, whereas staff may be more guarded when talking to patients or service users. It was found that 10% of staff had witnessed a colleague expressing the belief that someone could be ‘cured’ of their minority sexual orientation [3] despite the fact that conversion therapy has been condemned as ineffective [4] and that the UK government is bringing forward proposals to end the practice of ‘conversion therapy’ in the UK [5]. High proportions of health and care staff stated that they did not consider sexual orientation relevant to a person’s health needs, for example 72% of care workers, 62% of nurses, and 55% of social workers [3]. There is good evidence from the English General Practice Patient Survey [6] that sexual minorities have worse health care experiences than heterosexual people, which probably result in inequalities in family practitioner use [7]. From these results it is clear that health and social care staff need more effective equality and diversity training.

In 2006, Core Training Standards for Sexual Orientation was published with the aim to make national services inclusive for LGB people [8]. In 2010 the UK Parliament introduced the Equality Act (2010) which legally protects LGBT people from discrimination in the workplace and in wider society. The Unhealthy Attitudes Survey [3] found that only 50% of health and care staff reported that they had received equality and diversity training in the previous 12 months, and the majority of this did not address LGBT issues. Therefore, in spite of core training standards being available for 10 years, there continues to be a considerable need to increase the quality and effectiveness of equality and diversity training in the UK.

This systematic review evaluates all relevant materials about the delivery and evaluation of UK education of health and social care professionals in lesbian, gay, bisexual, and transgender (LGBT) issues, in order to obtain a starting point from which to improve training.

## 2. Materials and Methods

### 2.1. Inclusion Criteria

This systematic review was conducted according to a prospective protocol for a student project. Any qualitative or quantitative studies (published or grey literature) with information of interest in any setting and made available between 2010 and 2018 were eligible if they:Described evaluations of teaching to UK-based health and/or social care staff around LGBT issues; ofDescribed curricula or educational materials for use with UK-based health and/or social care staff around LGBT issues.

LGBT was defined as sexual orientation and gender identity minorities. Sexual orientation could be defined by identity (lesbian, bisexual, gay) or behaviour (women who described themselves as having sex with women, or having sex with women and men; men who described themselves as having sex with men, or having sex with women and men; or by same sex cohabitation status). Gender identity minorities were transgender men and women however defined. The 2010 cut off was chosen because of the UK Equality Act (2010).

Excluded were reports or studies calling for education to be improved, without giving any specific educational materials. Also excluded were reviews and systematic reviews of educational materials, organisational policy documents and materials from outside the UK.

### 2.2. Search Strategy, Study Selection, and Data Extraction

Database searches up to November 2019 were conducted by two reviewers (CB, CM) and checked by another (CM, RH). Databases (on platforms) searched included Applied Social Science Index and Abstracts (ProQuest), Embase (Ovid), Inform Adults (Community Care), Medline (EBSCO), PsychInfo (Ovid), Psychology and Behavioural Sciences Collection (EBSCO), Social Care Online (SCIE), Science and Social Science Citation Indices (Web of Science).

Search terms and appropriate synonyms (as MeSH terms and text words) were developed based on populations and exposures and included ‘sexual orientation’, ‘gender identity’ ‘gay’, ‘lesbian’, ‘bisexual’, ‘transgender’, ‘queer’, ‘questioning’, ‘cisgender’ ‘asexual’, ‘gender dysphoria’, ‘heteronormativity’, ‘men who have sex with men’ (MSM), ‘men who have sex with men and women (MSMW), ‘women who have sex with women’ (WSW), ‘women who have sex with men and women’ (WSWM), and ‘LGBT’ plus services and professionals involved such as ‘healthcare’, ‘social care’, ‘professionals’, and ‘practitioners’ plus terms around relevant LGBT issues such as ‘education’ ‘knowledge’, ‘understanding’, and ‘awareness’. The same search terms were used for each database but adapted where necessary. Database searches were supplemented with searches on Google, Google Scholar, and specific websites—the Stonewall charity, LGBT Partnership, LGBT Consortium, LGBT Foundation, Birmingham LGBT, ACCESSCare, LGBT Health and Wellbeing Scotland, MindOut, and London Friend. References of relevant reviews were sifted and the archives on LGBT health used in other projects by one of the authors (CM) was searched for relevant studies.

All titles found by the above search were assessed for inclusion and abstracts, where available, were read. If any titles and abstracts had relevant information or there was uncertainty, the full study was read and either accepted for the systematic review or rejected based on the above inclusion and exclusion criteria. Full-text assessment to determine inclusion in the systematic review was carried out by both reviewers (RH, CM). Any disagreements were resolved by discussion. Data was extracted by one reviewer (RH) and checked by another (CM). No authors were contacted about data discrepancies.

### 2.3. Data Analysis

Results are discussed narratively, with main themes developed through synthesis of qualitative results, and tabulation where appropriate. One researcher (RH) extracted relevant information from included studies, coded them, and organised them into descriptive themes. These were checked and amended by a second researcher (CM). None of the systematic review authors have been involved in any of the included studies.

## 3. Results

From the database searches 165 full papers were read. From these papers, 19 studies were included in the narrative synthesis, 3 in Group 1 and 16 in Group 2, see Figure 1 (PRISMA flow chart).

### 3.1. Study Characteristics

For Group 1 we found three action research projects evaluating teaching around LGBT issues to UK-based health and/or social care staff, one based in six residential care homes for older people [9,10], one with cancer nurses and other health professionals [11] and one with a local branch of a cervical screening programme in NHS Bradford and Airedale [12]. These are detailed in Table 1. There were no evaluations found for other NHS or social care staff training initiatives.

Action research involves research where the researchers and clients collaborate in the identification and definition of an issue to be addressed and in the development of solutions. All three action-research projects were successfully completed and reported important gains in understanding and attitudes in participants. For example, the residential care home project found at the 7-month-post-intervention interviews that there were small but important shifts in attitudes and gains in awareness. This then translated in more appropriate behaviour at key points. For the breast cancer project, it resulted in increased staff understanding of the distinctive needs of LGBT cancer service users, influencing of their attitudes and assumptions, the provision of tailored information and support from the two cancer charities involved, and wider dissemination through organisation staff members. For the cervical smear project open discussion of issues in training sessions led to successful countering of inaccurate views that might have hindered progress in the project.

For Group 2, we found 16 different sets of training materials around LGBT issues specifically for UK health and social care staff. These are listed in date order:Moving forward: working with and for older lesbians, gay men, bisexuals and transgendered people. Training and resource pack. Written by Steve Pugh, Willie McCartney, and Julia Ryan. (2010) [13]Working with older lesbian, gay, and bisexual people, a guide for care and support services. Written by James Taylor at Stonewall (2011) [14]Supporting older lesbian, gay, bisexual and transgender people, a checklist for social care providers. Written by Opening Doors London and Camden AgeUK (2011) [15]Implications for lesbian, gay, bisexual, and transgender (LGBT) people. www.scie.org.uk, written Social Care Institute for Excellence (SCIE) (2011) [16]Sexual Orientation: A guide for the NHS. Written by Alice Ashworth for Stonewall (undated but produced in 2012) [17]Working with lesbian, gay, bisexual, and transgender older people, by Trish Hafford-Letchfield (2014) [18]How to be LGBT+ friendly: Guide for care homes. Written by PrideCymru (2015) [19]LGB&T People & Mental Health: Guidance for Services and Practitioners. Written for the LGB&T Partnership by Barker MJ, et al., (2015) [20]Lesbian, gay, bisexual, trans and queer good practice guide. Mind and Mind Out (2016) [21]Dementia Care and LGBT communities: A good practice paper. Written by National LGBT Partnership and Colleagues (2016) [22]Out loud, LGBT voices in health and social care, a narrative account of LGBT needs. Written by LGBT Partnership (2016) [23]Best Practice in providing healthcare to lesbian, bisexual and other women who have sex with women. Written by LGBT Partnership (2016) [24]Lesbian, gay, bisexual & trans health priorities, building an LGB&T voice into planning systems. Written by LGBT Partnership (2017) [25]A whole systems approach to tackling inequalities in health for lesbian, gay, bisexual and trans (LGBT) people, a toolkit. Written by LGBT Partnership (2018) [26]Health4LGBTI Trainer’s Manual and 4 slide packs-Reducing Health Inequalities experienced by LGBTI People: What is Your Role as a Professional? Written by Zeeman and colleagues for the European Commission (2018) [27]Safe to be me. Meeting the needs of older lesbian, gay, bisexual and transgender people using health and social care services. A resource pack for professionals. Written by Sally Knocker and Anthony Smith for Age UK (undated but produced in 2018) [28]

### 3.2. Description of Documents for Group 2

For a brief description of the sixteen sets of training materials please see Table 2. 

#### 3.2.1. Material Recipients

Eleven of the documents were aimed at service managers, planners and/or commissioners. Of these, two [17,19] could have been easily accessed by individual staff to increase their knowledge and awareness. Two documents [14,27] were aimed at trainers and provided the tools to undertake the training of others. Four of the documents [16,18,20,21] were aimed primarily at front line staff and two were aimed at mental health practitioners and two were aimed at social care staff. In many cases, these documents outlined recommendations as to what should be covered in training for various groups of staff and the rationale for providing such training. Some documents were addressed to “anyone working or volunteering” in health and social care. Five documents were specifically aimed at health services only [17,20,21,25,26], six were for those providing health and social care [14,22,23,24,27,28], and five specifically at those providing social care [13,15,16,18,19]. Social care here is used in its widest sense including residential care and housing providers.

#### 3.2.2. Material Format

Only one of the documents contained ‘ready to use’ training materials [27]. Another provided all that might be required to design appropriate training [13]. Others provided plans of what materials to use dependent on the planned trainees and the level of contact—for example what would be appropriate for use with hospital porters and what would be appropriate in training those responsible for assessing a person’s care or health needs (for example SCIE 2011 [16]). Without exception the training materials or plans were for half day or whole day training events.

#### 3.2.3. Aims of Training Materials

In some instances, the aims of the training materials were stated directly, for example ensuring that the recipients knew what to do and what policies to implement in order to comply with the law (e.g., complying with Cree 2006 [8]). Others sought to “promote equality” or reduce health inequalities (for example LGB&T Partnership 2017 [25] and European Commission 2018 [27]). Some aimed to offer practical advice to improve services (for example Taylor 2011 [14], Pride Cymru 2015 [19] Barker 2015 [20] and Mind 2016 [21]) or provided checklists for evaluating current service provision (for example Opening Doors (London) 2011 [15] and Knocker and Smith 2018 [28]). Two documents (both of these were written for the NHS) also addressed the needs of LGBT staff and how they should be employed and supported [17,28].

#### 3.2.4. Specific Content

All the documents sought to give information in some manner, the aim being to influence recipients of the training to change behaviour. In some instances, this was directed towards those who worked directly with LGBT people as service users/patients as to how they should make services accessible inclusive and/or appropriate to the service user group. Others sought to influence managers in providing and evaluating training for their staff. Content included:
Use of language—Many items included glossaries, meanings of terms, what words to use and not use, how to avoid being exclusive (for example, by assuming heterosexuality) and offered specific examples of how to ask open questions in a non-exclusive manner. For example: “which people are important in your life?” [14] or “are you in a relationship?” [23] rather than assuming a heterosexual partner. Being seen to be prepared to challenge any homophobic remarks was also essential [28].Visual communication—Advice was given on how to promote an LGBT friendly ambience, including the use of pictures of same sex couples in health settings (for example Ashworth 2012 [17]) and in marketing, the use of rainbow images as a sort of kite mark (for example LGBT Partnership 2016a [22]), and the provision of LGBT specific magazines in waiting areas and residential facilities (for example Pride Cymru 2015 [19]) and the prominent display of policies on discrimination [14]. One gave examples of flags used in the community [21].Legal and policy position—That required by law was outlined (for example the Equalities Act, 2010; the Gender Recognition Act, 2004). Additionally, the expectations of professionals such as medical professionals (the NHS charter) and social workers (the Knowledge and Skills Framework) were explained and attempts were made to show how these might translate into practice for patients/service users. The organisation’s own policy statement was often explored with indications as to what should be done in order to comply.LGBT history—Some documents, particularly those aimed at individuals and organisations working with older people, explained what LGBT people’s life experience was likely to have been. The aim here was information giving but also so that training recipients could gain some insight into older LGBT people’s life history and expectations of discrimination when receiving health or care services.Checklists against which organisations and individuals could assess themselves were provided, together with examples of good practice: for example, Opening Doors (London) 2011 [15] and Knocker and Smith 2018 [28]. One provided an example of a monitoring form for sexual orientation and gender identity [21].Intersectionality was a common feature of the documents (for example Knocker and Smith 2018 [28]). It was frequently highlighted that the LGBT community was heterogeneous and that factors such as age, race, class, economic status, education all influenced the individual and their perspective and expectations of services (for example Pugh 2010 [13], Ashworth 2012 [17] Hafford-Letchfield 2014 [18])


Nine documents specifically concerned training/education with respect to older LGBT people (although of course some of these documents could apply to non-older people, for example people with early onset dementia, or in a care home due to a physical disability rather than due to age related issues). Three documents were specifically about trans people and five specifically excluded trans people unless they identified as lesbian, gay or bisexual, as the concern of these documents was sexual orientation rather than gender identity. Finally, 11 documents stated that they were including trans people, but not always with any specific content about specific trans needs.

#### 3.2.5. Pedagogical Methods

Pedagogical methods were varied. Only one document provided ready to use training materials that could be implemented alongside a facilitator’s handbook [27]. Another document provided the wherewithal to produce training by indicating which pages in the pack should be turned into PowerPoint slides and which exercises to use [13]. Among the documents, 11 provided case studies for discussion or examples of best practice (both individual and organisational best practice). Some provided tips or examples, for instance, how to ask open questions. Many documents used direct quotations from LGBT people as to their experience of how non-inclusivity made them feel or their experiences of discrimination within health and social care services (for example LGBT Partnership 2016b [23]).

## 4. Discussion

### 4.1. Main Findings

There have been no previous systematic reviews of UK education of health and social care professionals in LGBT issues or of evaluations of those training packages. Three action research projects were found which successfully addressed LGBT issues with some NHS cervical screening staff, breast cancer nurses, and private residential care home staff. The three projects in Group 1 were all participatory action research projects and these types of projects are known to have potential biases such as experimenter bias—a process where the researchers performing the research influence the results, in order to portray a certain outcome. However, participatory action research leads to co-production of outcomes with the clients so can have more insightful impact on the communities involved. In total, 16 training packages or sets of materials specifically targeting UK health and social care staff were found. Some of these training materials were from the same organisation or partnership but had different sources, were orientated towards different groups, e.g., dementia, health or health and social care, and contain different materials. The organisations were mostly LGBT-specific and so were very knowledgeable about the sector.

There are a number of position statements from UK health and care organisations, all addressing the need for training staff in health and social care [29,30,31,32,33,34]. Whilst these position statements draw attention to the inadequate state of current care, none of them offer further detail on how to improve experiences of LGBT patients and service users.

The issue for discussion here is that despite the availability of appropriate training programmes for post qualifying staff, recommendations to undertake training, best practice examples and statements of good intent, LGBT people continue to report that they are experiencing discrimination or direct prejudice from health and/or social care services. We must therefore ask the reasons as to why this might be the case.

As Peel (2007) cited in Westwood and Knocker (2016) [35] identified, “training stems from the belief that ‘negative attitudes and behaviours towards lesbians and gay men can be challenged through education”. A systematic review of studies evaluating how to change heterosexuals’ attitudes towards homosexuals found 17 empirical studies of mixed designs [36]. Most of the studies used educational interventions and/or contact with homosexuals to change heterosexuals’ prejudices. Careful analysis of the included studies listed in that systematic review suggests that interventions were effective if they involved direct interaction between the heterosexuals and a homosexual peer or lecturer that they already knew, and many of the purely educational interventions without personal interaction were ineffective. Also, emotive films such as “The Life and Times of Harvey Milk” were effective whereas a video depictive homosexual lifestyles and celebrating Gay Pride was not. Therefore, training materials incorporating more personalised attitudes and behaviour change techniques would be more likely to be effective with health and social care staff than the currently available materials.

Given the documents identified in this review and the expectation of training provision by organisations such as the Care Quality Commission (CQC), it would appear that training has not yet resulted in the desired outcomes. As we have seen, a lot of information is provided in the included documents, including legal responsibilities, organisational expectations, appropriate language to use and general awareness training concerning LGBT lives, and experience. Such training tends towards providing knowledge and, to a certain extent, related competencies rather than trainees’ abilities to employ emotional intelligence and to empathise with LGBT people. Oxman et al. (1995), reviewing the effectiveness of 102 educational interventions in health settings, question whether and how professional practice can be improved concluding that there are no ‘magic bullets’ [37]. They also conclude that there is a need to identify the reasons for sub-optimal performance and the barriers to change. In terms of changing practice in general practice, Wensing et al. (1998) concluded that interventions which simply employed knowledge transfer were less effective than interventions that also used social influence and management support; “knowledge transfer was necessary but insufficient to achieve change in practice routines” [38]. 

The LGBT training sessions recommended were all for periods of half or full day. Westwood and Knocker (2016) [35], when considering training to support those working with LGBT people who have a diagnosis of dementia, suggest that such training might become simply a tick box exercise such that managers can demonstrate to inspectors that staff have undertaken relevant training. There is relatively little evidence of training being evaluated, other than for the action research studies included in Group 1, and where evaluation is mentioned as having taken place this has tended to be at the end of the day of training rather than after time has elapsed. In this respect, any evaluation is likely to have a recency effect and it would be more useful explore whether trainees attitudes change following social immersion back with their peer group.

A frequent staff response to training such as this is that it is unnecessary as “we treat everyone the same”. Such an attitude demonstrates an inability to understand that treating everyone the same does not result in everyone receiving an equally good service. Person-centred treatment is cited throughout the documents as being essential, but in reality this will have little impact if staff have a poor understanding of the impact of interventions, treatments and ambience on LGBT individuals. The relevance of LGBT to general health issues (as opposed to sexual health) is not acknowledged. Similarly, there is acknowledgement within the documents that the LGBT community is not homogeneous and that a huge variety of other factors—such as race, ethnicity, class, gender, economic status, disability, etc.—all impact on the individual’s experience. It is unclear whether this emphasis on intersectionality as an important aspect of the interaction between professionals and service users/patients results in changes in practice.

### 4.2. Strengths and Limitations

Strengths include the development of a protocol, extensive searches for any relevant UK studies and inclusion of studies from a variety of sources, including grey literature. The main limitations are the difficulty of developing themes in this area from a variety of different types of training materials. We acknowledge that the thematic analysis is of a basic descriptive nature. Although every effort was made to find all relevant training materials some may have been missed. However, it is unlikely that any missed training materials would have been influential in the themes developed. We acknowledge that the included papers in Group 2 are not fully published and peer reviewed papers, but educational materials are rarely, if ever, published and peer reviewed. As grey literature, developed by committed activists or charities, they may be biased towards the LGBT sector but are made by people very knowledgeable about this sector. Also, they are aimed at staff education rather than promotion of research findings. Some researcher bias may have influenced our theme selection and their development, but our thematic analysis attempts to develop themes in an unbiased way as possible.

### 4.3. Implications for Policymakers

Many health and social care staff exhibit poor behaviour towards LGBT patients and service users. This is contrary to the expectations of the NHS Constitution and positions statements of relevant organisations. For example, the Royal College of Nursing is committed to reducing health exclusion and inequalities, challenging stigma, and unlawful discrimination in health care [34]. Guidelines exist on how expected behaviour of staff, but these do not seem to have been audited regarding LGBT issues. It is unclear how the excellent policy aims will be achieved, since there have been core training standards available since 2006 [8] but these do not seem to be used widely. Better training for health and social care staff is needed.

### 4.4. Implications for Research

There has been no research evaluating how best to encourage UK health and social care professionals to deliver appropriate care to LGBT patients or clients. Large, well conducted studies are needed to establish the effectiveness and appropriateness of current curricular developments such as the new training package for staff supporting young LGBTQ people in care [39]. Training materials incorporating more evidence-based attitude and behaviour change techniques should be developed and then evaluated to ensure their effectiveness with health and social care staff in a wide variety of settings.

## 5. Conclusions

Given that there is a wealth of resources available for training health and social care staff in the UK, some of which has been available for over a decade, it seems surprising that surveys such as the Unhealthy Attitudes Survey [3] and The National LGBT Survey [1] are still finding that some LGBT patients and service users face heteronormativity, inappropriate care, and occasional overt homophobia from health and social care staff. It is also worrying that, given that the materials are produced by very knowledgeable organisations, it is even more worrying that they do not appear to be positively influencing staff attitudes and behaviours. It seems evident that either the training packs that have been developed are not being used, or that they are being used but are not sufficiently effective. Training materials incorporating more evidence-based attitude and behaviour change techniques should be developed and evaluated properly. It is important for LGBT patients and service users to know that they will not face ignorance or hostility from any health and social care staff. Until staff are properly trained and aware of the issues, this will continue to occur.

## Figures and Tables

**Figure 1 ijerph-16-04976-f001:**
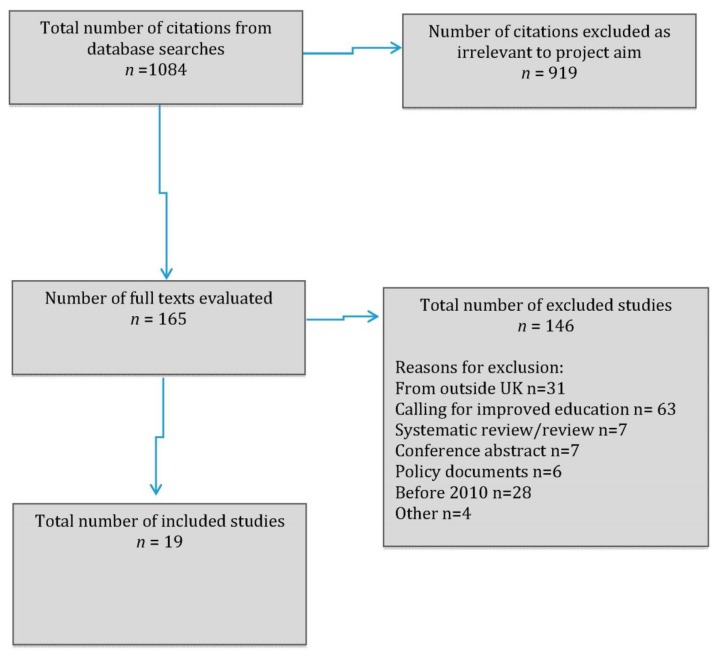
Prisma flow diagram.

**Table 1 ijerph-16-04976-t001:** Characteristics of included teaching project evaluations—Group 1.

	Service	Project Type	Number of Staff Involved	Number of Community Activists Involved	Actions	Findings	Follow Up	Grant Funding
Carter 2012	Cervical screening programme in Bradford and Airedale	Service improvement project for NHS	Not reported (?all staff in the cervical screening programme)	Not reported but some were in the project steering group	Materials for SMW developed with staff (leaflet) and circulated through GP practices, sexual health clinics, LGB venues, and Bradford’s gay pride event.	Being involved in update staff training was key to sustaining and widening impact	Evaluation of cervical screening rates in 2012 planned*	Department of Health Pacesetters initiative
Fish 2016	Cancer care	Knowledge exchange project	Approximately 9: Staff from a breast cancer charity, another cancer charity, a cancer research charity, NHS, 2 academics, and 5 cancer service users and carers	Staff from two LGBT community groups.	Funding applications, developing research questions and conducting research, contribution to National Cancer Equalities Initiative, staff seminars, good practice resources for staff, policy briefings, website material.	Wide ranging and successful project, resulted in lesbians and bisexual women being mentioned in policy statements	Not described specifically	National Cancer Action Team, ESRC Knowledge Exchange programme
Hafford-Letchfield 2017 and Willis 2018	Six residential care homes for older people in a large city in England	Service standard improvement for private care home provider	35 interviews (?all staff in each of the care homes)	8 (training was given to them)	Community activists co-facilitated staff advisory sessions then started dialogues with staff, with some difficult and important conversations.	Staff perceived sessions as enlightening, educational, and informative	Post-intervention interviews at 7 months	Comic Relief

* post intervention evaluation never occurred due to lack of funds (personal communication Lesley Hedges 2017).

**Table 2 ijerph-16-04976-t002:** Characteristics of training materials—Group 2.

	Author or Organisation	Target Group	Care Group Age	Target Provider	Length
1	Pugh 2010	LGBT	Older people	Health and social care	142 pages
2	Taylor 2011	LGB	Older people	Care and support services	28 pages
3	Opening Doors London 2011	LGBT	Older people	Social care providers	10 pages
4	SCIE 2011	LGBT	Personalisation	Social care providers	6 pages
5	Ashworth (2012)	LGB	All	Healthcare	23 pages
6	Hafford-Letchfield 2014	LGBT	Older people	Social care providers	30 pages
7	PrideCymru 2015	LGBT+	People in care homes	Care home providers	3 pages
8	Barker 2015	LGBT	All	Health services and practitioners	12 pages
9	Mind 2016	LGBTQ	All	Mental health service providers	23 pages
10	LGBT Partnership 2016a	LGBT	Dementia care	Dementia services	16 pages
11	LGBT Partnership 2016b	LGBT	All	Health and social care	24 pages
12	LGBT Partnership 2016c	SMW	All	Healthcare	22 pages
13	LGBT Partnership 2017	LGBT	All	Health and social care	14 pages
14	LGBT Partnership 2018	LGBT	All	Health systems	43 pages
15	European Commission (2018)	LGBT	All	Healthcare	Trainer’s manual 151 pages. Module 1–41 pages Module 2–61 pages Module 3–31 pages Module 4-41 pages
16	Knocker 2016-8	LGBT	Older people	Health and social care	40 pages

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
