# Peer review of "A Systematic Review of UK Educational and Training Materials Aimed at Health and Social Care Staff about Providing Appropriate Services for LGBT+ People"

_ijerph, 2019, doi:10.3390/ijerph16244976_

Round 1

Reviewer 1 Report

INTRODUCTION

The theoretical justification is limited.

MATERIALS AND METHODS

This section should be structured in two blocks: (1) Search, and (2) Document selection process. They should also present the description of the methodology in a more clear and orderly manner. For example, on lines 113-115 it says: "... the full study was read and either accepted for the systematic review or rejected based on the above inclusion and exclusion criteria". Exclusion criteria are not specified.

Figure 1 should reflect the entire document selection process, in detail. For example, when they say that a set of documents has been excluded, it is necessary that they make a breakdown of the total number indicating the reasons for exclusion in each case.

RESULTS

The results should be completed with quantitative data, for example, the number of publications per year, by theme, etc.

The presentation of results is disorganized and incomplete, in my opinion.

DISCUSSION AND CONCLUSIONS

Actually, they should emphasize the limitations of the study, the most important being that a review is made at a basic descriptive level.

Author Response

With a word count of over 4000, we felt that the introduction needed to be short and to the point, in the same way that many other systematic review introductions tend to be briefer rather than more extensive.

The order of the information in the methods section is the same as specified in the PRISMA checklist. This places the eligibility criteria first.

Thank you for pointing out that we had omitted the exclusion criteria – these are now added.

Reasons for exclusion of full texts is now in Figure 1.

The number of publications per year can be seen in the text on pages 9-12. The qualitative analysis gives references to studies reporting the various themes. Unlike the materials for most systematic reviews, the educational materials are by their very nature not quantitative so it is unclear how they can be described quantitatively in a way that would help the reader understand the themes prevalent across the teaching materials and in such a way as to draw out what is missing. The qualitative analysis by themes is, we agree, at a relatively basic level. We have added this as a limitation in the discussion section.

The only other systematic review of LGBT training materials we have found (Sekoni et al 2017), looked at mainly published studies from USA and looked at effects of the training on healthcare students and professionals, rather than looking at the contents in depth. This was described as a mixed methods systematic review and also has very little quantitative data.

Reviewer 2 Report

The article deals with an interesting topic, with important consequences both on the health and social level. Consequently, the methodology must be appropriate and compliant with the provisions of guidelines such as those developed by PRISMA. The analysis of the content highlights some critical issues in this regard. In general, the use of the so-called gray literature is not advisable in the context of a scientific review especially if this literature constitutes the absolute majority of the works examined. Another general critical point is the fact that many of the articles come from the same source or from connected sources. For example in group 2 five references are from the same organization and two other pairs are connected to it: the data should be highlighted and discussed in the light of the apparent poor results of the training activities on the subject highlighted. The greater dissatisfaction with health and social services can also be found in religious and ethnic minorities and even "economic", a comment on the specificity of the phenomenon described from line 40 to 45, also in light of what was said in lines 336-338. I also list a series of more specific observations (always in the PRISMA reference frame).
Row 47 heterosexual and cisgender are terms that define two sets one included in the other so for clarity only one of the two terms should be used.
Row 50 the survey was not conducted for scientific purposes, but in the context of the activities of an LGBT organization, this must be stated explicitly, since the rules for the validation of the questionnaire are not declared
. The comment reported on lines 56-57 has racist and anti-religious connotations despite other equally exemplar comments without these aspects are present in the document.
The "adaptation" in row 108 must be described, moreover the meaning of a search on google and Stonewall after thatal described in lines 95 to 99 is not clear.
Supposing that what is written from line 21 to 123 refers to the methodology commonly referable to the so-called "narrative medicine", the terms and classifications must be summarized in the table with their frequencies. (
If quantitative synthesis is not appropriate, describe the type of summary planned, PRISMA 15d) In figure 1 the scientific works must be separated and evidenced with respect to those coming from "gray literature" or pamphlet; furthermore, a summary of the reasons for inclusion / exclusion would be appropriate. see n° 8 PRISMA (Specify the study characteristics (e.g., PICO, study design, setting, time frame) and report characteristics (e.g., years considered, language, publication status) to be used as criteria for eligibility for the review) and N° 12 List and define all variables for which data will be sought (e.g., PICO items, funding sources), any pre-planned data assumptions and simplifications.

A better compliance with Prisma checklist 14 (Describe anticipated methods for assessing risk of bias of individual studies, including whether this will be done at the outcome or study level, or both; state how this information will be used in data synthesis) is also to be considered.

In the materials and methods section, you should enter how the strength of the body of evidence will be assessed (e.g., GRADE) or why it was not possible. Similarly any planned assessment of meta-bias (es) (e.g., publication bias across studies, selective reporting within studies) should be described. Finally, a point that would give greater emphasis to the discussion is the discussion about the fact that health personnel follow courses of study closely adhering to a biological model (not biopsychosocial) and in particular emphasis is placed on evidence based medicine. How much does the training material examined use "evidence based" examples or situations?

Author Response

We note the issue around use of grey literature but educational materials are rarely, if ever, published nor subject to peer review in the way that epidemiological studies are. We have included this as a limitation in the discussion section.

We agree that 5 of the sets of training materials appear to come from one source. However the LGBT Partnership is a partnership of ten organisations (see https://nationallgbtpartnership.org/about-the-partners/ ) so the writing of each of the five educational materials may have come from different sources, which is why they are orientated towards different groups, eg Dementia, Health, or Health and social care, and have different materials in them. We have added this point to the first part of the discussion section.

We agree that there is greater dissatisfaction with health and social services in religious and ethnic minorities and that is reprehensible. There are many similarities across disadvantaged groups, but we feel that that topic would be better in another paper.

You state that ‘heterosexual and cisgender are terms that define two sets one included in the other so for clarity only one of the two terms should be used’. This is incorrect. Heterosexual refers to sexual orientation and cisgender refers to gender identity. A person can be transgender and heterosexual and also cis-gender and lesbian. The two concepts are different and should not be confused.

Regarding Row 50 – the survey – I think you are referring to the Unhealthy Attitudes Survey by Stonewall? This survey was conducted using an online interview administered to members of the YouGov Plc. panel of 350,000+ individuals who have agreed to take part in surveys. So it wasn’t actually done by Stonewall. The report writing was done by Stonewall Equality Ltd, another organisation attached to the Stonewall Charity. With any project it is difficult to know whether vested interested might have influenced the findings, and this is true of drug company research as much as charity research. We can only declare the source of the information clearly so that the reader can make up their own mind about the veracity of the findings.

We have removed the quote about the nurse from Nigeria and replaced it with a similar homophobic quote without reference to nurses from abroad.

The adaptation of searches for different databases is normal for systematic reviews because similar concepts are indexed differently in the different databases and also if you use a different platform (Ovid, EBSCO etc). A systematic review usually has several means to find includeable studies, such as database searches, reference list checking, contact with experts and specific searches. Google and charity websites such as Stonewall were searched because, as pointed out above, educational materials are mostly not published so wouldn’t appear on publication databases such as Medline.

The PRISMA checklist was written for systematic reviews of epidemiological studies, primarily RCTs. It is not possible to adhere to every exact specification when the nature of the studies included in the systematic review are not epidemiological studies but, in this case educational materials. For example, there is no comparator in an educational material so the C of PICO does not apply. GRADE is for assessing outcomes across quantitative studies so again would not be relevant here. We have done what we can to describe the features of these materials as accurately and completely as we can in the narrative synthesis and in tables 1 and 2. 

Finally, you ask how much the educational materials use evidence-based examples – this is the whole point of this systematic review, the educational materials do not use the evidence about how to change peoples’ attitudes and behaviour.

Reviewer 3 Report

In their manuscript, Hunt et. al present a systematic review and synthesis on training and education of health and social care workers treating LGBT+ patients/clients in the UK. The paper is well written, and the topic is extremely important; however, further work needs to be done to describe the methodology, to include more recent studies, to globalize the data, and to flesh out the limitations.

Overall, the paper could use a good edit for English grammar and syntax (i.e.: missing commas, etc…) Some of the discussion becomes confusing as the authors talk about both provision of care and about workplace interactions in the introduction; however, they are attempting to discuss patient/client care throughout the rest of the manuscript. Please clarify and expand the dialogue throughout the study for workplace interactions or, perhaps, constrain the introduction to just the topic discussed in the manuscript. Why was the inclusion criteria limited to studies and grey literature after 2010? Were any important studies missed prior to the 2010 cutoff? The authors use a variation of LGBT+ throughout the manuscript, for a study such as this, it may be better to use the all inclusive “LGBT+” in all usages of the acronym or variations thereof. The overall search time frame ending in June 2018 seems woefully out-of-date. An updated search and synthesis should be conducted to include papers that have been published in the past year and a half since the searches were conducted. This is especially important with all of the attention given to this important topic in recent media. Why was the study limited to the UK? It seems there would be a lot more pertinent information if the study was expanded globally. Indeed, if the authors insist on keeping the study UK-specific, this fact should be directly reflected in the title of the manuscript. Lines 278-279: Please avoid statements of primacy The limitations, including all sources of bias, need to be more fully detailed in the “Strengths and Limitations” portion of the manuscript.

Author Response

We thank the peer reviewer for their initial comments. We would have liked to have made this review global but there was no funding for this project and the time it would have taken to amass training materials (published and in grey literature) would have been considerable, as would analysing the results. Also the environment that LGBT people find themselves in the different countries is very different, and health and social services are organised and funded differently than in the UK, so training materials will be differently focused. It would be necessary to analyse the results by country. This may be the topic of a future EU funding application.

We have combed the manuscript for grammatical and punctuation mistakes.

We have added an additional sentence in the introduction to explain that staff remarks to other staff have the potential to more clearly indicate their attitudes, whereas staff may be more guarded when talking to patients or service users.

We would welcome any recent UK studies or educational materials as we are unaware of any that have been made available since our search cut-off date.

We do not understand your comment about workplace interactions – the material in the introduction is there to explain why education of health and care staff on LGBT matters is important and needed, not to give a comprehensive review of workplace interactions.

We limited the materials to 2010 because of the Equalities Act (2010). A sentence on this is now in the introduction and the methods section.

Regarding the use of LGBT+ please see our comment above. This paper was on sexual orientation not gender identity so using the T in LGBT was not appropriate here. We have been precise in our use of abbreviations – for example the Cree Standards (2006) was only about sexual orientation, and the Equalities Act (2010) included gender identity.  

We can update the searches if required but it is unlikely that any new material found (if there is any) will change the core findings of the thematic synthesis.

We have included UK in the title now.

Unfortunately we haven’t been able to understand your comment about primary and as you don’t say which part of the manuscript this refers to we haven’t been able to act on it.

We have expanded the discussion on sources of bias in the discussion section. This includes discussion of biases in sources papers (in first part of discussion section) and in the review process (in the strengths and limitations section).

Round 2

Reviewer 1 Report

First of all, thank you for introducing some changes that were requested. I have also been able to verify that they have introduced other improvements in response to requests from other reviewers. Overall, I consider that the work has improved and that it responds to the objectives set by the authors.

Author Response

We thank the peer reviewer for their comments

Reviewer 2 Report

As for eterosexual SLASH cisgender: I'm sorry I wasn't clearer, so I repeat the concept with mathematical symbology. Given A = ethero ⋂ cisgender (person eterosexual and not LGBT), B = ethero ⋂ transgender (transgender and heterosexual), C = cisgender ⋂ LGB (fi cis-gender and lesbian) follows that except for the set A, cisgender ⋂ LGB therefore slash should not be used. Furthermore, based on the given definition of cisgender, the whole cisgender set is made up of several subsets, some of which intersect with the heterosexuals set and others with the LGB ones, but it would be incorrect to compare a set (cisgender) with some of its subsets since cisgender ⋂ LGB is not the empty set.

I agree that the focus of the work is not on minorities, but a reference to how widespread the phenomenon (negative experiences) is helps to understand if the topic of the work is specific or part of a general phenomenon.  Eventually it could also be a discussion point  regarding the results of the training interventions (is the failure "specific" for LGBT trainig or common to all minorities; are the educational modalities the same?).

The materials object of the study may be hardly present on the scientific databases, however it is necessary to specify more clearly whether the search on google scholar has used the same terms (probably even a search on google could be used to identify other material "not scientifically published" so why limit to the scholar version?). Finally if other websites have been explored beyond Stonewall they must be indicated, otherwise the phrase "specific websites such as .." is incorrect; a clarification on what is meant by "specific website" is necessary to better understand the width of the research.

in row 370 "very knowlegeable" needs a reference.

The narrative analysis summarized in Tables 1 and 2 must be briefly described: have textual analysis programs been used (as normally in narrative medicine) and if so, which ones? Have the key concepts been extrapolated on the basis of their recurrence or a theoretical scheme constructed a priori has been used?

"The educational materials do not use evidence on how to change peoples attitudes and behavior." This is not the evidence to which I referred: the health education paradigm provides that only theories and practices supported by scientific evidence are taught (eg it is shown that vaccination prevents infection), this also applies to non-strictly biological aspects: for instance EBM-based sanitary courses teach that the collaboration of the assisted person and an empathic attitude of the professionals may improve the results of the therapy. Are there any scientific elements in the training material to support the gender theory or to contrast statements such as "we treat everyone the same" (row 349) or "I know only X and Y chromosomes"? This is an important point in the discussion that cannot be limited to only hoping for a greater use of "evidence based attitude and behavioral change tecniques" (by the way bct are widely used in conversion therapy, see row 73 for its efficacy).

Lines 402-404 are not a consequence of what was said in the introduction or of the analyzed data, therefore they do not belong to the conclusion and can be eliminated or moved to the beginning of the discussion.

Since most, if not the totality of training materials comes from sources connected to the LGBT community, it is appropriate to consider this factor in the discussion: it can be considered a facilitating element or a barrier (according to the ICF terminology) for overcoming the prejudice. In figure 1 the reasons for the exclusion of the 695 citations must be inserted. Moreover in the same figure the results of the search on database or on google scholar must be indicated separately.

Author Response

We have numbered the peer reviewer's comments for clarity of response

1. Heterosexual is not the same as cisgender. We do not understand your relational algebra. We have removed the / and replaced with ‘and’.

2. We are experts in LGBT health and care, not in inequalities in health and care. We do not have the expertise to be able to say whether similar negative experiences are experienced by other minorities when accessing healthcare.

3. We have added details of other websites searched. We did use Google and Google Scholar, this is now added.

4. It seems self-evident that LGBT organizations are knowledgeable about LGBT issues. We do not think this needs a reference.

5. We have already described our narrative analysis. We think the peer reviewer is describing automated text analysis which is not appropriate for this type of synthesis.

6. We do not understand the peer review comment here. Also to suggest that conversion ‘therapy’ is efficacious is wrong and insulting. Please see the UK Government  LGBT Action Plan 2018, see here: https://www.gov.uk/government/publications/lgbt-action-plan-2018-improving-the-lives-of-lesbian-gay-bisexual-and-transgender-people

7. We disagree – these lines sum up the issues raised in this paper.

8. We have added a sentence in the discussion about materials coming from the same or similar sources. In systematic reviews it is not expected that reasons for exclusions at title and abstract phase are given, this is only for the full text assessment. It would be inappropriate to waste time in this way and it seems evident that this inappropriate request is designed to delay or frustrate publication. We do not have the information of the number of citations from the different sources as they were all entered into one file. This information is not usually requested for systematic review publications.

Reviewer 3 Report

Thank you for considering the majority of my previous comments. Two issues are still outstanding:

1) Please perform further searches for more material published post mid-2018 as it is now a year and a half past the search cut off date.

2) Please define whether the paper has focus on dealing with LGBT+ staff in the workplace and interactions between coworkers OR if the paper is about LGBT+ individuals/patients seeking care and their experiences in the system.

Author Response

1. We have updated the searches and updated the PRISMA diagram. We found no new evaluations for group 1 and two extra educational materials for group 2. These have now been inserted into the manuscript results.

2. The paper is not about any of your suggested topics but about training of healthcare professionals. Comments or quotes by staff about LGBT staff are included to show the level of awareness of staff to LGBT concerns, and therefore the need for their education. Education of managers in the needs of LGBT staff has also been included.

Round 3

Reviewer 2 Report

Point 2 As mentioned in the previous report, I do not ask for an analysis of the problem, but a sentence that helps to realize how the situation of the LGBT community is quantitatively relevant or qualitatively different from potentially similar situations (to exemplify the type of reference see Racial / Ethnic Disparities in the Assessment and Treatment of Pain. Psychosocial Perspectives Saint Louis University School of Medicine) Point 5 Narrative analysis has well-defined criteria (see f.i. Charon R, Wyer P. Narrative evidence based medicine. Lancet 2008; 391: 296–7. ) which are not sufficiently described in the text. Evidently there is a problem of communications: in point 6 the comment in brackets was not aimed at an alleged efficacy of the conversion theraphy, on the contrary it emphasized that the ineffectiveness of the techniques used could pose doubts about their effectiveness also in other situations. The use in support of a government report quotation when there are numerous scientific publications available seems to me to be inappropriate. The authors seem to believe that the problem is quantitative: we need to do more intensely than we are already doing. The hypothesis (perhaps to deny it) that can be qualitative is not taken into consideration: what is done is effective, corresponds to what the subjects of the intervention are used to? For point 7 I also disagree: the two final sentences are not fully substained by what is highlighted in the review (see point 6). I consider the insinuation in paragraph eight to be offensive and I do not agree with what was stated on the exclusion at the title and abstract level (relevance, duplicates, poor quality, …) the exclusion criteria are not irrelevant: any elimination of potential information sources must be transparent to eliminate the suspicion of involuntary bias.

Author Response

Point 2. As we explained in our previous response, we are not experts in comparison of the LGBT community to other communities such as racial/ethic communities in the UK. Our article is about the LGBT community issues, not a comparison between different minority groups. So we do not plan to make any changes.

Point 5. We thank the peer reviewer for the reference and we enjoyed the painting by James McNeill Whistler in the article. The narrative medicine described there is to do with physician experience as compared to evidence-based medicine. There are no defined criteria in the article. See https://www.thelancet.com/article/S0140-6736(08)60156-7/fulltext

We agree that there has been miscommunication regarding conversion therapy and it is important for peer reviewers to be clear in the points that they raise. The government policy document used in support of our rebuttal has a summary of the scientific evidence in it. As it is in the rebuttal not the paper, comments about its appropriateness will not improve the paper.

We do not understand the comment about the problem being quantitative.

Point 7 – since the other peer reviewers are happy with the paper we do not propose to make any changes, as we disagree with this peer reviewer.

Point 8. We repeat that the justification of elimination of citations at titles and abstracts stage is not expected by the PRISMA criteria, I have never seen it done and I have never had to do it any of my ~150 published systematic reviews.

Reviewer 3 Report

The authors have addressed my concerns. Thank you.

Author Response

We thank the peer reviewer